

**Moment magnitude estimates for Central Anatolian earthquakes using coda waves**
Tuna Eken[1]
[1]*Department of Geophysical Engineering, the Faculty of Mines, Istanbul Technical*
*University, 34469 Maslak, Sarıyer, Istanbul, Turkey*
*Abstract*
Proper estimate of moment magnitude that is a physical measure of the energy released at
earthquake source is essential for better seismic hazard assessments in tectonically active
regions. Here a coda wave modeling approach that enables the source displacement spectrum
modeling of examined event was used to estimate moment magnitude of central Anatolia
earthquakes. To achieve this aim, three component waveforms of local earthquakes with
magnitudes $2.0 \leq M_L \leq 5.2$ recorded at 72 seismic stations which have been operated
between 2013 and 2015 within the framework of the CD-CAT passive seismic experiment.
An inversion on the coda wave traces of each selected single event in our database was
performed in five different frequency bands between 0.75 and 12 Hz. Our resultant moment
magnitudes ($M_W$-coda) exhibit a good agreement with routinely reported local magnitude
($M_L$) estimates for study area. Finally, we present an empirical relation between $M_W$-coda and
$M_L$ for central Anatolian earthquakes.
Keyword(s): Coda waves modelling, seismic moment, moment magnitude, Radiative Transfer
Theory



### 1. Introduction

The robust and stable knowledge of source properties (e.g. moment magnitude estimates) is
crucial in seismically active countries such as Turkey for a better evaluation of seismic hazard
potential as this highly depends on establishment of reliable seismicity catalogs. Moreover,
accurate information on source parameters could be important when developing regional
attenuation properties.

Conventional type of magnitude scales ($M_L$, $m_b$, $M_S$) as the result of empirically derived using
direct wave analyses can be biased due to various effects such as source radiation pattern,
directivity, and heterogeneities along the path since they may cause drastic changes in direct
wave amplitude measurements (e.g., Favreau and Archuleta, 2003). Instead several early
studies depending on the analysis of local and/or regional coda envelopes have indicated that
coda wave amplitudes are significantly less variable by a factor of 3-to-5 compared to direct
wave amplitudes (e.g., Mayeda and Walter, 1996; Mayeda et al., 2003; Eken et al., 2004;
Malagnini et al., 2004; Gök et al., 2016). In fact local or regional coda waves that are usually
considered to be generally to be composed of scattered waves and can be simply explained by
that sample the single scattering model of Aki (1969) have been proven to be virtually
insensitive to any source radiation pattern effect in contrast to direct waves because of the
volume averaging property of the coda waves sampling the entire focal sphere (e.g., Aki and
Chouet, 1975; Rautian and Khalturin, 1978). In Sato and Fehler (1998) and Sato et al. (2012)
an extensive review study on the theoretical background of coda generation and advances of
empirical observations and modelling efforts can be found in details.

There have been several approaches used for extracting information on earthquake source size
via coda wave analyses. These approaches can be mainly divided into two groups. The first





group of studies employs coda normalization strategy in which measurements require a
correction for seismic attenuation parameters (e.g. intrinsic and scattering) that can be
described by some empirical quality factors. To calibrate final source properties reference
events are used to adjust measurements with respect to each other. For forward generation of
synthetic coda envelopes, either single-backscattering or more advanced multiple-
backscattering approximation are used. An example to this group is an empirical method
originally developed by Mayeda et al. (2003) to investigate seismic source parameters such as
energy, moment, and apparent stress drop in the western United States and in Middle East.
They corrected observed coda envelopes for various influences, for instance, path effect, S-to-
coda transfer function, site effect, and any distance-dependent changes in coda envelope
shape. Empirical coda envelope method have been successfully applied to different regions
with complicated tectonics such as northern Italy (e.g. Morasca et al., 2008), Turkey and
Middle East (e.g. Eken et al., 2004; Gök et al. 2016); or Korean Peninsula (e.g. Yoo et al.,

64  2013).


Second type of approach is a joint inversion technique that is based on a simultaneous
optimization of source, path, and site specific terms via synthetic and observed coda envelope
fitting within a selected time window including observed coda and direct-S wave parts. In this
approach, the Radiative Transfer Theory (RTT) is employed for analytic expression of
synthetic coda wave envelopes. The method that does not rely on coda normalization strategy
was originally developed by Sens-Schönfelder and Wegler (2006) and successfully tested on
local and regional earthquakes ($4 \leq Ml \leq 6$) detected by the German Regional Seismic
Network. Further it has been applied to investigate source and frequency dependent
attenuation properties of different geological settings, i.e., Upper Rhine Graben and Molasse
Basin regions in Germany and western Bohemia/Vogtland in Czechia  (Eulenfeld and Wegler,



2016); entire United States (2017); central and western North Anatolian Fault Zone (Gaebler
et al., 2018; Izgi et al., 2018). A more realistic earth model in which anisotropic scattering
conditions were earlier considered by Gusev and Abubakirov (1987) yielded peak broadening
effects of the direct seismic wave arrivals. This approach later was used in previous studies
(e.g. Zeng, 1993; Przybilla and Korn, 2008; Gaebler et al., 2015) that dealt with propagation
of P-wave elastic energy and the effect of conversion between P- and S-wave energies.

In the current work I present estimated source spectra as an output of a joint inversion of S-
and coda waves parts of local earthquake waveforms 487 local earthquakes with magnitudes
$2.0 < ML < 4.5$ detected in central Anatolia for their source parameters. The approach used
here employs isotropic acoustic RTT approach for forward calculation of synthetic coda
envelopes. Gaebler et al. (2015) has observed that modeling results from isotropic scattering
were almost comparable with those inferred from relatively more complex elastic RTT
simulations with anisotropic scattering conditions. The use of a joint inversion technique is
advantageous since it is insensitive to any potential bias, which could be introduced by
external information, i.e., source properties of a reference that is obtained separately from
other methods for calibration. This is mainly because of the fact that we utilize an analytical
expression of physical model involving source, and path related parameters to describe the
scattering process. Moreover the type of optimization during joint inversion enables the
estimates for source parameters of relatively small sized events compared to the one used in
coda-normalization methods.






*2. Regional Setting and Data*
Present tectonic setting of Anatolia and surrounding regions have been mainly outcome of the
northward converging movements among Africa, Arab, and Eurasian plates. To the west
subducting African plate with a slab roll-back dynamics beneath Anatolia along Hellenic
Trench has led to back-arc extension in the Aegean and western Anatolia while compressional
deformation to the east around the Bitlis–Zagros suture was explained by collisional tectonics
(e.g. Taymaz et al., 1990; Bozkurt, 2001). Westward extrusion of Anatolian plate controlled
by these plate motions, in consequence, has been accommodated through two conjugate
strike-slip fault zones separating the Anatolian and Arabian from Eurasian plates: 1600 km
long east-west striking transform plate boundary, North Anatolian fault zone (NAFZ), and
northeast-southwest–striking East Anatolian fault zone (EAFZ) (Fig. 1). These neotectonic
features could have easily traced the weakness zones along the boundaries of amalgamated
continental fragments that have developed following the closure of Tethys Ocean (Şengör et
al., 2005).

Central Anatolia is located between extensional regime to the west due to the subduction and
compressional regime tectonics to the east due to the collisional tectonics. The major fault
zone in the region, the Central Anatolian Fault Zone (CAFZ) (Fig. 2), which primarily
represents a transtensional fault structure with small amount of left-lateral offset during the
Miocene (e.g. Koçyiğit and Beyhan, 1998), can be considered as a boundary between the
carbonate nappes of the Anatolide-Tauride block from the highly deformed and
metamorphosed rocks in the Kırşehir block. However recent studies that have reported
significant lateral variations in seismic wave speeds (e.g. Fichtner et al., 2013a,b; Delph et al.,
2015) and Bouguer gravity  (Ateş et al., 1999) across the fault implied that a progressive
relative movement along the faults would result in sharp difference in crust and mantle



structures. New findings on structural, geomorphic, and geochronologic data collected from
several segments along the CAFZ were interpreted that the transtensional type deformation
has reactivated paleotectonic structures and finally accommodated E-W extension due to the
westward extrusion of Anatolia (Higgins et al., 2015). To the northwest of the CAFZ, Tuz
Gölü Fault Zone (TGFZ) (Fig. 2), which is characterized by a right-lateral strike slip motion
with a significant oblique-slip normal component, appears to be collocated with Tuz Gölü
Basin sedimentary deposits as well as crystalline rocks within Kırşehir Block (e.g. Çemen et
al., 1999; Bozkurt et al., 2001; Taymaz et al., 2004). Present day crustal deformation and state
of stress related to the TGFZ have been reported in Çubuk et al. (2014) via observed
earthquake cluster activity reaching depths of 5-6 km with magnitudes up to $M_L 5.6$ in the
Bala region (between 2005 and 2007) located at the north of the TGFZ (Çubuk et al., 2014).

At the southwest tip of the study region, the EAFZ generates large seismic activity that can be
identified rather complicated seismotectonic setting: predominantly left-lateral strike-slip
motion correlated well with the regional deformation pattern but also existing local clusters of
thrust and normal faulting events on NS- and EW-trending subsidiary faults, respectively
(Bulut et al., 2012). Such complicated behavior explains kinematic models of the shear
deformation zone evolution. This active left-lateral fault zone since the late Miocene–Pliocene
exhibits ~20 km-wide shear deformation zone with an annual 6-10 mm/yr slip rate. It
connects to the NAFZ at the Karlıova Triple Junction (Bozkurt, 2001) and to the south splits
into various segments nearby the Adana Basin (Kaymakci et al., 2006) (Fig. 2). Toward the
south, the EAFZ reaches the Dead Sea Fault Zone (DSFZ) that has a key role in
accommodating northward relative motions of Arabian and African Plates with respect to
Eurasia.

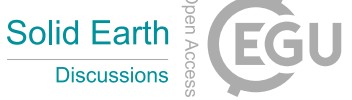

The present work utilizes three-component waveforms of local seismic activity detected at 72
broadband seismic stations (Fig. 2) that have been operated for 2 years between 2013 and
2015 within the framework of a temporary passive seismic experiment, the Continental
Dynamics–Central Anatolian Tectonics (CD-CAT) (Portner et al., 2018). We benefit from
revisited    standard    earthquake    catalogue    information    (publicly    available    at
http://www.koeri.boun.edu.tr) to extract waveform data for a total of 2231 examined events
with station-event pair distance less than 120 km and focal depths less than 10 km. Most of
the detected seismic activity in the study area is associated to several fault zones in the region,
i.e., the EAFZ, CAFZ, DSFZ, TGFZ, etc. Here we note that selection of only local
earthquakes is to exclude possible biases, which may be introduced by Moho boundary
guided Sn-waves while upper crustal earthquakes are preferred in this study to exclude effect
of relatively large-scale heterogeneities on coda wave trains. Finally a visual inspection
conducted over all waveforms to ensure high-quality waveforms reduces our event number to
1193. Selected station and event distributions can be seen in Figure 2.

Observed waveforms were prepared at 5 different frequency bands with central frequencies at
0.75, 1.5, 3.0, 6.0, 12.0 Hz via a Butterworth band-pass filtering process. In the next step, we
applied Hilbert transform to filtered waveform data in order to obtain the total energy
envelopes. An average crustal velocity model was used to predict P and S wave onsets on
envelopes and then based on this information: (i) the noise level prior to the P-wave onset was
eliminated (ii) S-wave window was determined starting at 3s prior to and 7 s afterwards S-
wave onset as this allowed to include all direct S-wave energy, (iii) starting at the end of the
S-wave window, a coda window of 100s at maximum was determined. Length of coda
windows can be shorter when signal-to-noise ratio (SNR) is less than 2.5 or when the same





window consists of coda waves from two earthquakes, which can give rise to a decline in the
envelope. We omit the earthquakes with less than 10 s of coda length from our database.

*3. Method*
We adopted an inversion procedure that was originally developed by Sens-Schönfelder and
Wegler (2006) and later modified by Eulenfeld and Wegler (2016). The forward part, which
involves calculation of energy density for a specific frequency band caused by an isotropic
source, is expressed in Sens-Schönfelder and Wegler (2006) as follows:
$$E_{mod}(t, r) = WR(r)G(t, r, g)e^{-bt} \quad (1)$$

*where* $G(t, r, g)$ represents Green's function that includes scattered wave field as well as
direct wave. W gives source term and it is frequency dependent. R(r) indicates the energy site
amplification factor and b is intrinsic attenuation parameter.
Possible discrepancy between predicted and observed energy densities for each event at each
station with $N_{ij}$ time samples (index k) in a specific frequency band can be minimized using:

$$\epsilon(g) = \sum_{i,j,k}^{N_S, N_S, N_{ij}} \left( lnE_{ijk}^{obs} - lnE_{ijk}^{mod}(g) \right)^2 \quad (2)$$

Here, the number of stations (index i) and events (index j) are shown by $N_S$ and $N_E$,
respectively. Optimization of g will be achieved when

$$lnE_{ijk}^{obs} = lnE_{ijk}^{mod} \quad (3) \quad \text{or}$$

$$lnE_{ijk}^{obs} = \ln G t_{ijk}, r_{ijk}, g + lnR_i + lnW_j - bt_{ijk} \quad (4)$$



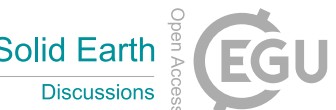

Equation 4 simply define an overdetermined inversion problem with $\sum_{i,j} N_{ij}$ number equation
systems and with $N_S + N_E + 1$ variables and thus $b$, $R_i$, and $W_j$ can be solved via a least-
squares technique. $\epsilon(g)$ can be defined as sum over the squared residuals of the solution.
Eulenfeld and Wegler (2016) present a simple recipe to perform inversion:
(i) Calculate Green's functions through the analytic approximation of the solution for 3-D
isotropic radiative transfer (e.g. Paasschens 1997; Sens-Schönfelder and Wegler, 2006) by
using fixed scattering parameters and minimize equation 4 to solve for $b$, $R_i$, and $W_j$ via a
weighted least-squares approach.
(ii) Calculate $\epsilon(g)$ using equation 2.
(iii) Repeat (i) and (ii) by selecting different $g$ to find the optimal parameters $g$, $b$, $R_i$ and $W_j$
that finally minimize the error function $\epsilon$.
In Fig. 3 an example for the minimization process that was applied at five different frequency
band is displayed for one selected event at recorded stations of the CD-CAT project.
Minimization described above for different frequencies will yield unknown spectral source
energy term, $W_j$ as well as site response, $R_i$ and attenuation parameters, b, and g. The present
study deals with frequency dependency of $W_j$ since this information can be later useful to
obtain source displacement spectrum and thus seismic moment and moment magnitudes of
analyzed earthquakes using the formula of the *S*-wave source displacement spectrum for a
double-couple source in the far-field, which is given by Sato et al. (2012):
$$\omega M(f) = \sqrt{\frac{5\rho_0 v_0^5 W}{2\pi f^2}} \quad (5)$$





The relation between the obtained source displacement spectrum and seismic moment value
was earlier described in Abercrombie (1995) by:

$$\omega M(f) = M_0 \left(1 + \left(\frac{f}{f_c}\right)^{\gamma n}\right)^{-\frac{1}{\gamma}} \quad (6)$$

where n is related to the high-frequency fall-off and $\gamma$ is known as shape parameter that
controls the sharpness of spectrum at corner frequency between the constant level $M_0$ (low
frequency part) and the fall-off with $f^{-n}$ (high frequency part). Taking logarithm of equation 6
gives:

$$\ln \omega M(f) = \ln M_0 - \frac{1}{\gamma}\ln \left(1 + \left(\frac{f}{f_c}\right)^{\gamma c}\right) \quad (7)$$


Eq.7 describes an optimization problem of which data forms observed source displacement
spectrum and four source parameters, $M_0, \gamma$, n, and $f_c$ are the unknown model parameters that
can be resolved in a simultaneous least-squares inversion of the equation 7. Finally moment
magnitude, $M_W$ can be calculated from modeled source parameters, seismic moment, $M_0$
using a formula given by Hanks and Kanamori (1979):

$$M_w = \frac{2}{3} \log_{10} M_0 - 6.07 \quad (8)$$


*4.  Results and Discussions*
*4.1 Coda wave source spectra*
Figure 4 displays observed values of source spectra  established by inserting inverted spectral
source energy term W at each frequency in Eq. 5 for all analyzed events. Each curve in this
figure represents model spectrum estimate based on inversion procedure described in previous




section. Modeled spectrum characteristics computed for 487 local earthquakes of which
lateral distribution is presented in Figure 2 suggest, in general, that we were able to obtain
typically expected source displacement spectrum with a flat region around the low frequency
limit and decaying behaviour  above a corner frequency.

Owing to the multiple-scattering process within small scale heterogeneities that makes coda
waves gain an averaging nature, the variation in coda amplitudes due to differences source
radiation pattern and path effect are reduced (Walter et al., 1995; Mayeda et al., 2003).
Eulenfeld and Wegler (2016) found that radiation pattern would have only a minor influence
on the S-wave coda while it might disturb attenuation models inferred from the direct S-wave
analyses unless the station distribution relative to the earthquakes  indicates a good azimuthal
coverage. A peak-like source function assumption for small earthkquakes that are utilized in
the present work was earlier proven to be adequate in early application of the coda-wave
fitting studies (e.g. Sens-Schönfelder and Wegler, 2006; Gaebler et al., 2015; and Eulenfeld
and Wegler, 2016).

Conventional approaches (e.g. Abercrombie, 1995; Kwiatek et al., 2011) to estimate source
parameters such as corner frequency, seismic moment, high-frequency fall-off through fitting
of observed displacement spectra observed at a given station in an inversion scheme could be
misleading since these methods usually: (i) assume a constant value of attenuation effect (no
frequency variation) defined by a factor $\exp(-\pi ft Q^{-1})$ over the spectrum, (ii) and assume
omega-square model with a constant high-frequency fall-off parameter, n=2. Following Sens-
Schönfelder and Wegler (2006) and Eulenfeld and Wegler (2016), however, we estimate
attenuation parameters (intrinsic and scattering) seperately within a simultaneous inversion
procedure in which high-frequency fall-off parameter varies. This is fairly consistent with





early studies (e.g. Ambeh and Fairhead, 1991; Eulenfeld and Wegler, 2016) where significant
deviations from the omega square model (n>3) were reported implying that the omega-square
model as a source model for small earthquakes must be reconsidered in its general
acceptance. In our case, the smallest event was with $M_W$-coda larger than 2.0, thus we had no
chance to make a similar comparison, however, high-frequency fall-off parameters varied
from n=0.5 to n=4. A notable observation in the distribution of n was n=2 or n=2.5 would be
better explain earthquakes with $M_W$-coda >4.0 whereas the smaller magnitudes exhibited
more scattered pattern of variation in n. Eulenfeld and Wegler (2016) claimed that the use of
separate estimates of the attenuation or correction for path effect via emprically determined
Green's function would be better strategy in order to invert station displacement spectra for
source parameters. This is mainly because smaller earthquakes (with n>2), in particular,
assuming omega-square model can distort the estimates of corner frequency and even seismic
moment especially in regions where Q is strongly frequency dependent.

*4.2 Coda wave –derived magnitude vs. $M_L$ catalogue magnitude*
A scatter plot between catalogue magnitudes based on local magnitudes ($M_L$) and our coda-
derived magnitudes ($M_W$-coda) that are inferred from resultant frequency dependent source
displacement spectra and thus seismic moment (e.g. Eq. 8) is shown in Fig. 6. Such
comparison suggests an overall coherency between both types of magnitudes. This implies
very simple model of a first-order approximation for S-wave scattering with isotropic acoustic
radiative transfer approach can be efficient to link the amplitude and decaying character of
coda wave envelopes to the seismic moment of the source.


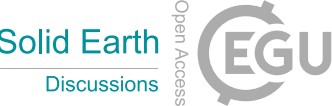

A linear regression analyses performed between $M_W$-coda and $M_L$ magnitudes (Fig. 5)
resulted in an emprical formula that can be employed to convert local magnitudes into coda-
derived moment magnitude calculation of local earthquakes in this region:

$$M_{W-coda} = 1.1655 \pm 0.0337 \times M_L - 0.7085 \pm 0.0128 \quad (9)$$

Apparent move-out in Fig. 5 and Eq. 9, presumably stems from the use of different magnitude
scales for comparison. The consistency between coda-derived moment magnitude and local
magnitude scales for the earthquakes with $M_W$-coda > 3.0 indicates that our non-empirical
approach successfully worked in this tectonically complex region. We observed similar type
of consistency in early studies that investigate source properties of local and regional
earthquakes based on emprical coda methods with simple 1-D radially symmetric path
correction (e.g. Eken et al., 2004; Gök et al., 2016). Observable outliers in Figure 5, for the
events with less than Mw 3.5, however, can be attributed to the either possible biases on local
magnitude values taken from the catalogue or small biases on our intrinsic ($Q_i^{-1}$) and
scattering ( $Q_s^{-1}$) attenuation terms. One another possible contribution to such mismatch might
be associated to the influences of mode conversions between body and surface waves or
surface-to-surface wave scattering (e.g. Wu & Aki 1985) that are not restricted to low
frequencies (<1Hz) (Sens-Schönfelder and Wegler, 2006).

*5. Conclusions*
This study provides an independent solution for estimating seismic source parameters such as
seismic moment and moment magnitude for local earthquakes in central Anatolia without
requiring *a priori* information on reference events with waveform modelling results to be
used for calibration or *a priori* information on attenuation for path effect corrections. In this



regard, the approach used here can be easy and useful tool for investigation of source
properties of local events detected at temporal seismic networks. Moreover, seismic moment
can be approximated via waveform modelling methods but due to the small-scale
heterogeneities of the media that waves propagate, it is often a hard task to establish Green's
function for small earthquake ($M_L < 3.5$). An analytical expression of energy density Green's
function in a statistical manner employed in the present work enables neglecting the
interaction of the small-scale inhomogeneities with seismic waves as this can be practical for
seismic moment calculations of small events that may pose source energy at high-frequency.
It is noteworthy to mention that our isotropic scattering assumption does not consider
anisotropic case, which could be valid for real media, but still provides a simple and effective
tool to define the transport for the anisotropic case since the estimated scattering coefficient
can be interpreted as transport scattering coefficient. An averaging over S-wave window
enables to overcome biases caused by using unrealistic Green's function (Gaebler *et al.*
2015). Since the present study mainly focuses on source properties of local earthquakes in the
study area, scattering and intrinsic attenuation properties that are other products of our coda
envelope fitting procedure will be examined in details within a future work. Finally, the
empirical relation developed between $M_W$-coda and $M_L$ will be a useful tool for quickly
converting catalogue magnitudes to moment magnitudes for local earthqukes in the study
area.

*Data and resources*
The python code used for carrying out the inverse modeling is available under the permissive
MIT license and is distributed at https://github.com/trichter/qopen. We are grateful to the IRIS
Data Management Center for maintaining, archiving and making the continuous broadband
data used in this study open to the international scientific community.




*Acknowledgement*
The facilities of IRIS Data Services, and specifically the IRIS Data Management Center, were
used for access to waveforms, related metadata, and/or derived products used in this study.
IRIS Data Services are funded through the Seismological Facilities for the Advancement of
Geoscience and EarthScope (SAGE) Proposal of the National Science Foundation under
Cooperative    Agreement    EAR-1261681.    Data    for    the    CD-CAT    experiment
(https://doi.org/10.7914/SN/YB_2013) are available from the IRIS Data Management Center
at http://www.iris.edu/hq/. Tuna Eken acknowledge financial support from Alexander von
Humboldt Foundation (AvH) towards computational and peripherals resources.

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



*Figure Captions*

Figure 1. Major tectonic features of Turkey and its adjacent. The plate boundary data used here is taken from Bird (2003). Subduction zones are black, continental transform faults are red, continental rift boundaries are green, and spreading ridges boundaries are yellow. NAFZ, EAFZ, and DSFZ are the North Anatolian Fault, East Anatolian Fault, and the Dead Sea fault, respectively.

Figure 2. Epicentral distribution of all local events selected from the study area in the KOERI catalogue. Gray circles represent earthquakes with poor quality that are not considered for the current study while black indicates the location of local events with good quality. Red circles among these events are 487 events used in coda wave inversion since they are successful at passing quality criteria of further pre-processing procedure.

Figure 3. An example from the inversion procedure explained in chapter 3. Here coda envelope fitting optimization is performed on band-pass filtered (8-16Hz) digital recordings of an earthquake (2014 April 09, $M_W$-coda3.2) extracted for 7 seismic stations that operated within the CD-CAT array. Large panel at the lower left-hand side displays the error function $\varepsilon$ as a function of $g_0$. Thick blue cross here represent the optimal value of $g = g_0$. Other small panels at upper and right-hand side show the least- squares solution of the weighted linear equation system for the first 6 guesses and optimal guess for $g_0$. There dots and gray curves indicate the ratio between energy ($E^{obs}$) and the Green's function (G) obtained for direct S-waves and observed envelopes at various stations, respectively. Please notice that during this optimization process envelopes are corrected for the obtained site corrections $R_i$. The slope of linear curve at each small panel yields $-b$ and while its intercept W are the intrinsic





attenuation and source related terms at the right-hand side of equation 4 part of the right-hand
side of the equation system.

Figure 4. a) Results of the inversion of the 2014-April-09, $M_W$-coda3.2 earthquake: Sample
fits between observed and calculated energy densities in the frequency band 0.5–1.0 Hz are
given for 6 different stations (see upper right corner for event ID, station name, and distance
to hypocenter). Note that light blue curves represent observed envelope. Smoothed observed
calculated envelopes in each panel are presented by blue and red curves, respectively. Blue
and red dots exhibit location of the average value for observed and calculated envelopes
within the S-wave window, respectively. b) The same as in (a) obtained in the frequency band
4.0–8.0 Hz.

Figure 5. All individual observed (black squares) and predicted (gray curve) source
displacement spectra observed at 72 stations from 487 local earthquakes in central Anatolia.

Figure 6: Scatter plot between local magnitudes ($M_L$) of analyzed events with coda waves-
derived magnitudes ($M_W$-coda) of the same events. The outcome of a linear regresssion
analysis yielded an emprical formula (e.g. Eq. 9) to identify the overall agreement represented
by gray straight line. Yellow and red dashed lines indicate upper and lower limit of linearly
fitting to that scatter.

Figure 7: Same scatter plot displayed in Fig. 6 color coded by estimated high-frequency fall-
off parameter for each inverted event.


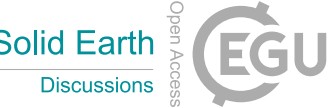





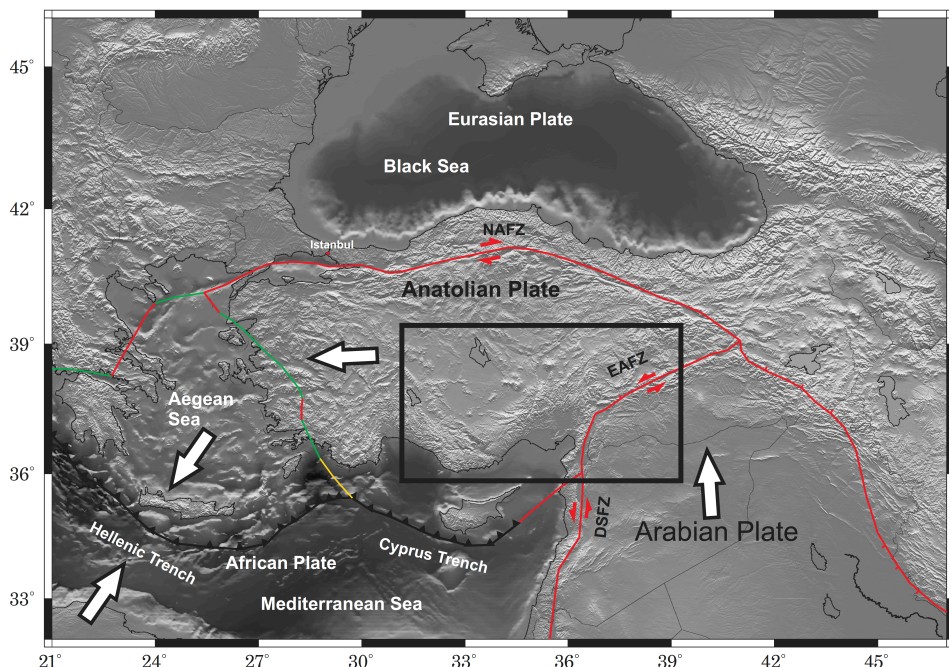


Figure 1.








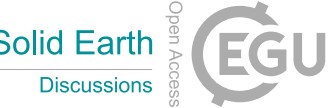




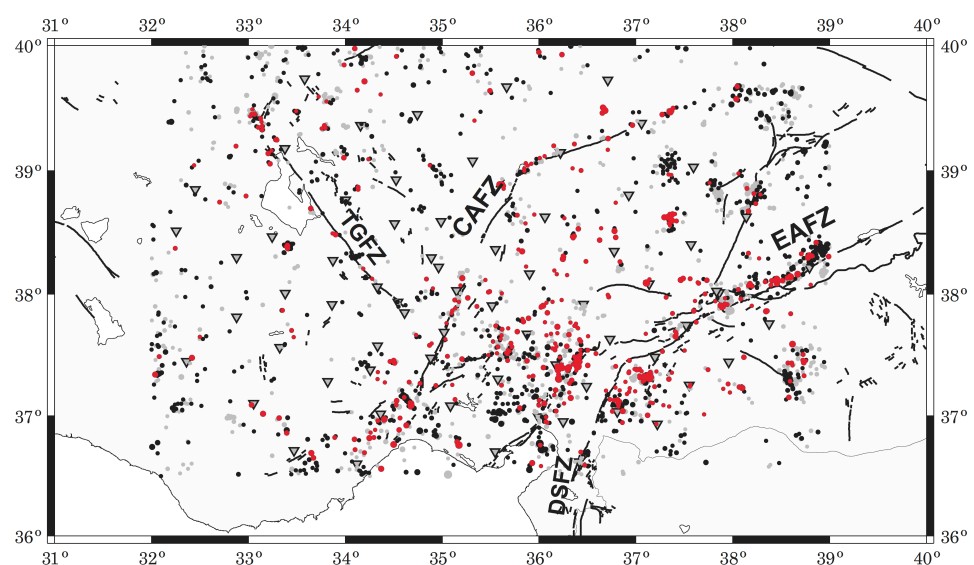


Figure 2.





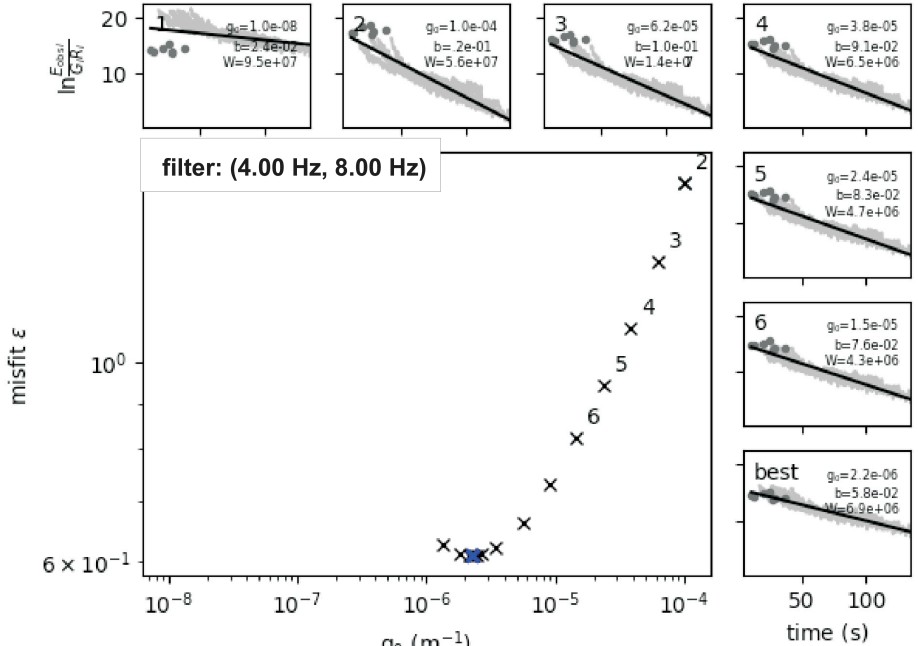

Figure 3.





Figure 4.





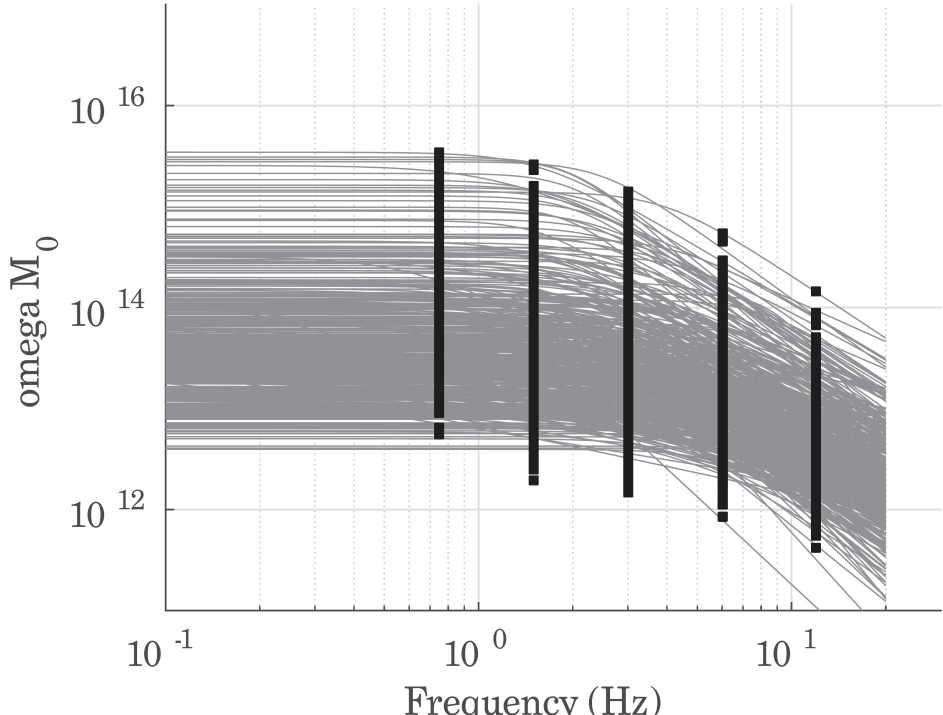

Figure 5.






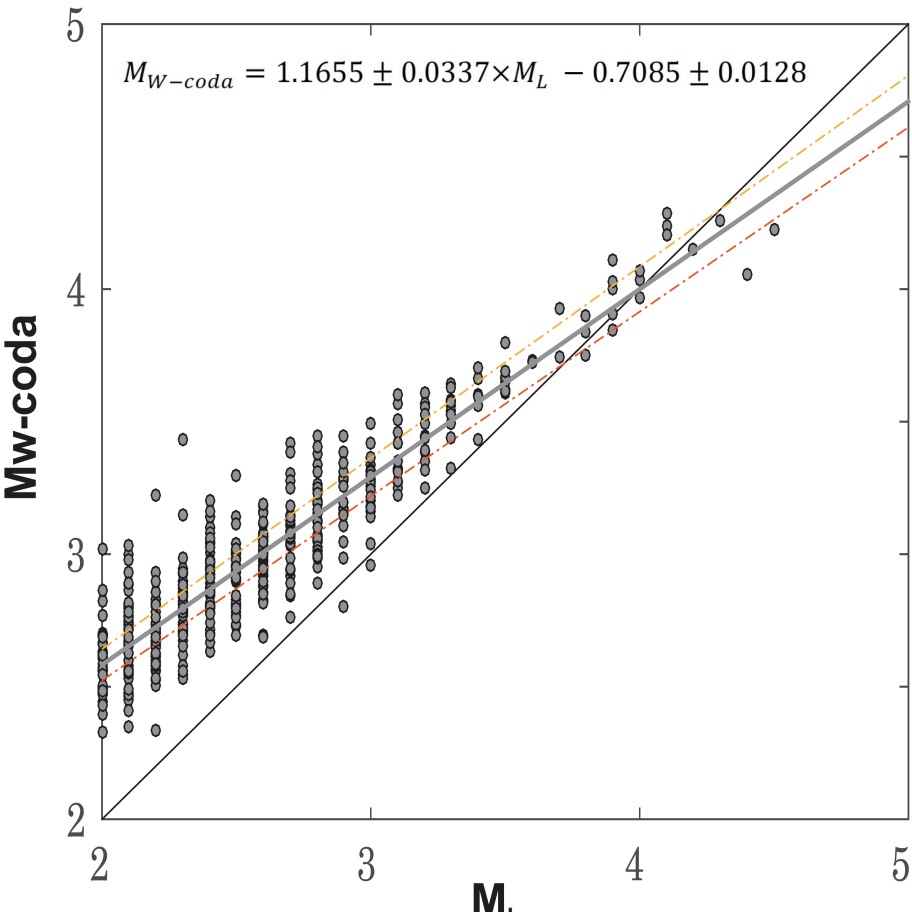

$$M_{W-coda} = 1.1655 \pm 0.0337 \times M_L - 0.7085 \pm 0.0128$$

Figure 6.




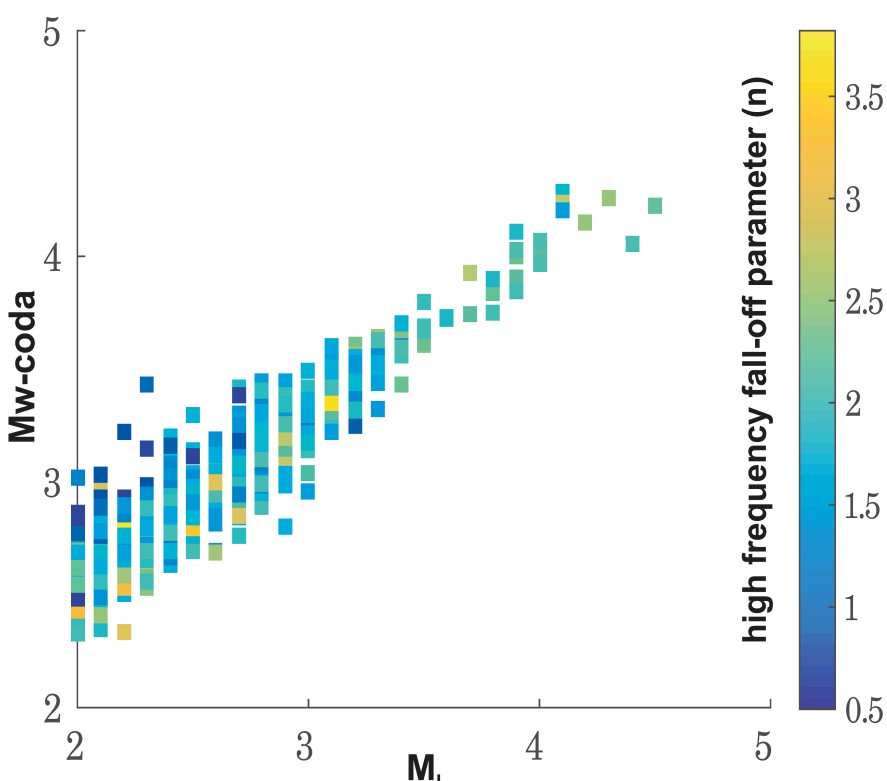

Figure 7.