# Peer review of "Moment magnitude estimates for Central Anatolian earthquakes using coda waves"

_Solid Earth, 2019_

## Referee Comment (RC1) · Takahiko Uchide (Referee) · 21 Feb 2019

The manuscript titled "Moment magnitude estimates for Central Anatolian earthquakes using coda waves" by Tuna Eken addresses the earthquake source characteristics revealed by decomposing the path and source effects in seismic data. Unfortunately, the manuscript is not well prepared: Description of the method is insufficient; some of figures are not correctly cited or never cited; a long review of geology is, however, not related to the discussion; and the discussion lacks some of important studies. It is hard to judge the value of this study from the present form of the manuscript. Detailed comments are listed below:

Specific Comments: 1. Section 2 (Reginal Setting and Data) has a long description

[Figure]

on the geological setting for four paragraphs, however, these are not related to any of discussions. It is nice to review the geological setting, but this can be significantly shortened for improving the readability.

2. Section 3 (Method) lacks an explanation on the g parameter. If possible, describe the formula of G as a function of g explicitly, so that it becomes understandable why the Author later used the grid-search scheme for optimizing the g parameter.

3. Figure 4 is cited in L239, however, Figure 4 and the text in L239-244 are inconsistent. Presumably Figure 5 should be cited here. Then Figure 4 is not cited anywhere. Figure 4 looks related to the estimation of the b parameter. Please clarify this.

4. The paragraph in L258-279 describes the demerit of the assumption of the frequency-independent attenuation factor and the omega-square source model, however, this paragraph should cite other studies already considering these problems. For example, Ide and Beroza (2001, doi: 10.1029/2001GL013106) pointed out the advantage of the empirical Green's function approach and corrected other studies' results one by one. As for the source spectra, Denolle and Shearer (2016, doi: 10.1002/2016JB013105) and Uchide and Imanishi (2016, doi: 10.1785/0120150322) pointed out the deviation of observed source spectra from the conventional omega-square model. Update the discussion by citing these papers.

5. The comparison between the coda-derived moment magnitudes and the local magnitudes was done simply done by the linear regression (equation (9)), however, it has been pointed out that moment magnitudes and local magnitudes are systematically different. Some of papers on this are Bakun and Lindh (1977, BSSA), Edwards et al (2009, doi: 10.1785/0120080292), Goertz-Allmann et al. (2011, doi:10.1785/0120100291), Munafo et al. (2016, doi: 10.1785/0120160130), Malagnini and Munafo (2018, doi: 10.1785/0120170303), and Uchide and Imanishi (2018, doi: 10.1002/2017JB014697). They proposed various types of regression curves, for example, composed of two straight line or a polynomial. Update the discussion by citing

these papers and correct the abstract (L17-19) accordingly.

6. Figure 7 is not cited anywhere. Add text related to Figure 7, or delete this figure.

Technical Comments: L253: earthkquakes -> earthquakes

————————————————————

---

## Author Comment (AC1) · 27 Feb 2019

Dear Editor,

I appreciate you for your interests and consideration of the present submitted study on investigation of source properties of central Anatolian earthquake using modelling of coda waves.

Here I would like to notice that the comments and opinions from Dr. Takahiko Uchide was very thorough and constructive . I carefully examined his comments and implemented necessary changes and corrections to the manuscript. I believe that his comments and suggestions substantially improved the quality and clarity of the paper.

I attach a file named "EKEN-SE-2019-8-REVISION-for-Reviewer1.pdf" as supplemen-

tary. This file contains the rebuttal letter, and main text with annotations showing any changes (in red color) and final version of the main text after implemented modifications.

Sincerely yours

Tuna Eken

Please also note the supplement to this comment:
https://www.solid-earth-discuss.net/se-2019-8/se-2019-8-AC1-supplement.pdf

**Supplement:**

**Responses to reviewer 1**
* * *
**Manuscript No.: SE-2019-8**
* * *
Dear reviewer,

I appreciate you for taking the interest and time to make this very thorough and constructive review on the present manuscript. I carefully studied your comments and made necessary changes and corrections to the manuscript. I hope our changes and corrections are sufficient to make our article suitable for publication soon. Your comments and suggestions certainly helped to improve quality and clarity of the paper.

I reply your comments below and highlight the changes (in red color) in the main text according to your suggestions.

Sincerely yours

Tuna Eken

**Reviewer 1**: The manuscript titled "Moment magnitude estimates for Central Anatolian earthquakes using coda waves" by Tuna Eken addresses the earthquake source characteristics revealed by decomposing the path and source effects in seismic data. Unfortunately, the manuscript is not well prepared: Description of the method is insufficient; some of figures are not correctly cited or never cited; a long review of geology is, however, not related to the discussion; and the discussion lacks some of important studies. It is hard to judge the value of this study from the present form of the manuscript. Detailed comments are listed below:

**T.E.:** I appreciate reviewer 1 for careful reading. I find the general points that reviewer 1 complains here fairly meaningful. Here in the modified text I considered all critical issues raised by reviewer 1 in details. In that respect I am thankful him since he pointed out deficit of the manuscript in a constructive manner as he suggested solutions to improve current shape of manuscript.

**Specific Comments**

**Reviewer 1**: Section 2 (Reginal Setting and Data) has a long description on the geological setting for four paragraphs, however, these are not related to any of discussions. It is nice to review the geological setting, but this can be significantly shortened for improving the readability.

**T.E.:** I perfectly understand the concern of reviewer 1 and I accordingly shortened Geological part by avoiding some details. The current version of the *Regional Setting* is now made much brief by mainly focusing around tectonic structures responsible observed seismicity that form main database of this study.

**Reviewer 1**: Section 3 (Method) lacks an explanation on the g parameter. If possible, describe the formula of G as a function of g explicitly, so that it becomes understandable why the Author later used the grid-search scheme for optimizing the g parameter.

**T.E.:** I appreciate this comment and agree that I missed analytical expression of Green's function. Now I took care of this and added mathematical expression (Now Equation 2 in the revised text) into the text.

**Reviewer 1**: Figure 4 is cited in L239, however, Figure 4 and the text in L239-244 are inconsistent. Presumably Figure 5 should be cited here. Then Figure 4 is not cited anywhere. Figure 4 looks related to the estimation of the b parameter. Please clarify this.

**T.E.:** Yes we appreciate reviewer 1. I made a mistake when citing Figure 5. We corrected this. Figure 4 is now also cited properly in the revised version of the text.

**Reviewer 1**: The paragraph in L258-279 describes the demerit of the assumption of the frequency-independent attenuation factor and the omega-square source model, how- ever, this paragraph should cite other studies already considering these problems. For example, Ide and Beroza (2001, doi: 10.1029/2001GL013106) pointed out the advantage of the empirical Green's function approach and corrected other studies' results one by one. As for the source spectra, Denolle and Shearer (2016, doi: 10.1002/2016JB013105) and Uchide and Imanishi (2016, doi: 10.1785/0120150322) pointed out the deviation of observed source spectra from the conventional omega- square model. Update the discussion by citing these papers.

**T.E.:** I appreciate this helpful comment and modified relevant part of the manuscript by discussing early works from other parts of the world that have reported deviations from the omega-square model with remarkable similarities to current findings from central Anatolia in this study. Added studies in the *Discussion* section is also updated in *Reference* section.

**Reviewer 1**: The comparison between the coda-derived moment magnitudes and the local magnitudes was done simply done by the linear regression (equation (9)), however, it has been pointed out that moment magnitudes and local magnitudes are sys- tematically different. Some of papers on this are Bakun and Lindh (1977, BSSA), Edwards et al (2009, doi: 10.1785/0120080292), Goertz-Allmann et al. (2011, doi:10.1785/0120100291), Munafo et al. (2016, doi: 10.1785/0120160130), Malagnini and Munafo (2018, doi: 10.1785/0120170303), and Uchide and Imanishi (2018, doi: 10.1002/2017JB014697). They proposed various types of regression curves, for example, composed of two straight line or a polynomial. Update the discussion by citing these papers and correct the abstract (L17-19) accordingly.

**T.E.:** I appreciate reviewer 1 for making me aware of some early studies in which non-linear form of regression strategy was utilized. I have added the details from those works into Discussion section regarding the comparison between MW-coda and $M_L$ catalogue magnitudes. There I also added a paragraph why we should not expect a perfect match between these two magnitude scales.

[revised manuscript text omitted]

---

## Referee Comment (RC2) · Takahiko Uchide (Referee) · 6 Mar 2019

On the revised manuscript titled "Moment magnitude estimates for Central Anatolian earthquakes using coda waves" by Tuna Eken, the Author sincerely responded to the Reviewer's comments. After correcting a very minor point (see below), this will be ready for the publication.

Text: L329: Add "(2018)" after "Malagnini and Munafò"
* * *

---

## Author Comment (AC2) · 6 Mar 2019

Dear Editor,

I am glad to see that Dr Uchide has found the article satisfactory that is very encouraging. I have now implemented the correction regarding very minor comment by Dr Uchide that is related to the missing year information of a given reference in the text.

I attach a file named "EKEN-SE-2019-8-REVISION2-for-Reviewer1.pdf" as supplemen- C1

Sincerely yours,

Tuna Eken

[Figure]

Please also note the supplement to this comment:
https://www.solid-earth-discuss.net/se-2019-8/se-2019-8-AC2-supplement.pdf

---

## Referee Comment (RC3) · Ludovic Margerin (Referee) · 16 Apr 2019

In this work, the author uses a multiple-scattering approach to infer the source spectra of small to moderate earthquakes recorded in central Anatolia using the observed energy envelopes of seismograms. The method, originally developed by Sens-Schoenfelder and Wegler (2006), is based on isotropic, scalar radiative transfer theory and makes use of a generalized inversion technique. The author obtains source spectra that are generally well fitted by the classical omega-squared source spectrum for the largest one. For smaller events, there is considerably more scatter in the exponent of the spectral decay at high frequency. The author shows that there exists a reasonably good correspondance between the local Magnitude ML and the coda-derived moment magnitude. I think that the study confirms the overall interest in using coda waves to

study the source of small earthquakes. It also suggests that the physics of smaller events might be different from the one of larger earthquakes. The analysis is sound and the study will be useful to convert local magnitudes to moment magnitudes in future investigations of the area. Therefore, I support the publication of the manuscript after the following questions/points have been addressed.

1) In introduction, I would suggest to distinguish more clearly between parametric approaches (such as the one developed by Mayeda and co-workers) and physics-based approaches (Wegler and co-workers). In the present version, what distinguishes the two methods is not really clear.

2) It is written that the method of Wegler does not rely on coda normalization. I think that this is an overstatement: although the authors do no explicitly "coda-normalize" their data, I still think that the separation of source and site effects still relies heavily on the fact that at long lapse-time in the coda, the energy distribution homogenizes spatially. If I am mistaken, please explain why.

3) I would recommend to split section 2 into two sections: Geology on the one hand and Data on the other hand

4) In the data section, it would be useful to state the final number of utilized paths after applying the selection criteria for the coda

5) In the Method section, there are a few typos in the Eqs, please verify. I would recommend to explain how g is updated in the inversion process. Furthermore, I think that it would be useful to discuss the possible trade-offs among the unknown in the system of Eqs (4)

6) In the studied area, I imagine that there are strong lateral variations of geology and therefore that the scattering coefficient depends on the source station pair. Is this taken into account in your inversion? If so, could you comment on the spatial variations of g in the studied area.

[Figure]

7) Is Figure 4 discussed somewhere in the text? I could not find. If the Figure is not useful, you should suppress it. If it contains information, please discuss it more carefully.

8) On L.250 and elsewhere, it is said that the radiation pattern has only a minor influence on the coda, an assertion with which I agree on the whole. Nevertheless, the separation of scattering and absorption also uses the energy contained the coherent wave which is strongly affected by the radiation pattern. If you have used techniques such as MLTWA in the past, you have probably observed that the largest fluctuations in the data occur in the window containing the ballistic wave. Therefore, it is not completely clear to me how the radiation pattern affects the data inversion. Could you comment on this point?

I also upload an annotated manuscript where I have made additional remarks, mostly pertaining to grammar and/or typos, but also to the presentation of Figures.

Please also note the supplement to this comment:
https://www.solid-earth-discuss.net/se-2019-8/se-2019-8-RC3-supplement.pdf

[Figure]

**Supplement:**

[revised manuscript text omitted]

*English*

*Figure 6 and 7 might be merged*

[Figure]

**Fig. 1**

[Figure]

**Fig. 2**

[Figure]

**Fig. 3**

[Figure]

**Fig. 4**

[Figure]

**Fig. 5**

[Figure]

$$M_{W-coda} = 1.1655 \pm 0.0337 \times M_L - 0.7085 \pm 0.0128$$

**Fig. 6**

[Figure]

**Fig. 7**

---

## Author Comment (AC3) · 26 Apr 2019

Dear Editor,

I appreciate you for your interests and consideration of the present submitted study on investigation of source properties of central Anatolian earthquake using modelling of coda waves.

Here I carefully went through the comments and questions from Dr. Ludovic Margerin, which I found very helpful and constructive in improving the manuscript. In addition, I have modified the manuscript, particularly Introduction, Results and Discussions Session, and Conclusion depending on the manuscript he prepared with his annotations for corrections. I corrected several typo and sentences with his helps.

[Figure]

Overall I carefully examined his comments and implemented necessary changes and corrections to the manuscript. I believe that his comments and suggestions substantially improved the quality and clarity of the paper. I should also notice here that I worked on his comments and suggestions for corrections on already revised version of the manuscript after review from Dr. Takahiko Uchide.

I attach a file named "EKEN-SE-2019-8-REVISED-REVIEWER2-ALL.pdf" as supplementary. This file contains the rebuttal letter, and main text with annotations showing any changes (in red color) and final version of the main text after implemented modifications.

Sincerely yours Tuna Eken

Please also note the supplement to this comment:
https://www.solid-earth-discuss.net/se-2019-8/se-2019-8-AC3-supplement.pdf

―――――――――――――――――

**Supplement:**

**Responses to reviewer 2**
* * *
**Manuscript No.: SE-2019-8**
* * *
**Reviewer 2:** In this work, the author uses a multiple-scattering approach to infer the source spectra of small to moderate earthquakes recorded in central Anatolia using the observed energy envelopes of seismograms. The method, originally developed by Sens-Schoenfelder and Wegler (2006), is based on isotropic, scalar radiative transfer theory and makes use of a generalized inversion technique. The author obtains source spectra that are generally well fitted by the classical omega-squared source spectrum for the largest one. For smaller events, there is considerably more scatter in the exponent of the spectral decay at high frequency. The author shows that there exists a reasonably good correspondence between the local Magnitude $M_L$ and the coda-derived moment magnitude. I think that the study confirms the overall interest in using coda waves to study the source of small earthquakes. It also suggests that the physics of smaller events might be different from the one of larger earthquakes. The analysis is sound and the study will be useful to convert local magnitudes to moment magnitudes in future investigations of the area. Therefore, I support the publication of the manuscript after the following questions/points have been addressed.

**T.E:** I appreciate for your very detailed reading and for positive encouraging opinion now about this study. The annotated text that you provided as supplementary figure among your revision process was too helpful in improving the text since I could correct several typos, rephrase unclear statements in this way. Below you can find my opinion corresponding to the issues you have raised and necessary corrections based on your comments. I have highlighted all changes in red color in the main text.

**Reviewer 2:** In introduction, I would suggest to distinguish more clearly between parametric approaches (such as the one developed by Mayeda and co-workers) and physics-based approaches (Wegler and co-workers). In the present version, what distinguishes the two methods is not really clear.

**T.E.:** We are thankful the reviewer for his careful reading this part. We can understand unclear points and performed some modification to enhance this part of the text.

**Reviewer 2:** It is written that the method of Wegler does not rely on coda normalization. I think that this is an overstatement: although the authors do no explicitly "coda-normalize" their data, I still think that the separation of source and site effects still relies heavily on the fact that at long lapse-time in the coda, the energy distribution homogenizes spatially. If I am mistaken, please explain why.

**T.E.:** Reviewer is partly correct because usually coda-normalization may fail for smaller events since the use of shorter coda waves n this case does not satisfy the requirement of homogeneous distribution of energy in space. This is mainly due to the fact that the presence of random seismic noise can dominate coda part. We avoid such shortcoming by involving source excitation and site amplification terms directly in the inversion process.

**Reviewer 2**: I would recommend to split section 2 into two sections: Geology on the one hand and Data on the other hand.

**T.E.:** I implemented this suggestion. Additionally, based on the suggestion of reviewer 1, I shortened the geology section was by leaving out redundant amount of knowledge in this section.

**Reviewer 2**: In the data section, it would be useful to state the final number of utilized paths after applying the selection criteria for the coda.

**T.E.:** This information is now added into the Data section.

**Reviewer 2**: In the Method section, there are a few typos in the Eqs, please verify. I would recommend to explain how g is updated in the inversion process. Furthermore, I think that it would be useful to discuss the possible trade-offs among the unknown in the system of Eqs (4).

**T.E.:** I thank reviewer 2 for his highlighting a few typos in related equations. They are corrected now. In a response to the reviewer #1, I have added analytical expression of Green's function to clarify its dependence on the scattering attenuation parameter g. I explain the stepwise procedure about how the parameter g is updated between lines # 212-218 (new manuscript). However, I should also notice that preferred to give a summarized version of the inversion strategy in this manuscript and tried to avoid going into too many details about attenuation parameter estimation in the manuscript mainly since the current work is just an application of a coda modeling approach previously developed by Sens-Schönfelder and Wegler (2006) and later modified by Eulenfeld and Wegler (2016) and secondly the main focus of the current work is on the source parameter estimation. I also added an explanation about possible trade-offs (lines# 206-210).

**Reviewer 2:** In the studied area, I imagine that there are strong lateral variations of geology and therefore that the scattering coefficient depends on the source station pair. Is this taken into account in your inversion? If so, could you comment on the spatial variations of g in the studied area?

**T.E:** We appreciate reviewer for raising this point. The reviewer 2 is right about lateral heterogeneities in the study region. To consider this I specifically classified station-event pairs into two regional groups: those lying within Kırşehir Block and within Anatolide-

Tauride Block that are separated by the Central Anatolia Fault System. However, I should also notice that the approach used here does not require a priori knowledge of scattering and intrinsic attenuation. Resultant attenuation terms estimated following the simultaneous inversion procedure indicated an overall dominancy of intrinsic attenuation over scattering one. A detailed discussion will be the out of our scope in the present study but certainly will be subject to future work that will be primarily focusing on crustal heterogeneities based on lateral variation of anelastic attenuation properties of the study region.

**Reviewer 2**: Is Figure 4 discussed somewhere in the text? I could not find. If the Figure is not useful, you should suppress it. If it contains information, please discuss it more carefully.

**T.E.:** This issue was earlier pointed out by reviewer 1 and it stems from a mistake when citing Figure 5. I already corrected this. Figure 4 is now also cited properly in the revised version of the text.

**Reviewer 2**: On L.250 and elsewhere, it is said that the radiation pattern has only a minor influence on the coda, an assertion with which I agree on the whole. Nevertheless, the separation of scattering and absorption also uses the energy contained the coherent wave which is strongly affected by the radiation pattern. If you have used techniques such as MLTWA in the past, you have probably observed that the largest fluctuations in the data occur in the window containing the ballistic wave. Therefore, it is not completely clear to me how the radiation pattern affects the data inversion. Could you comment on this point?

**T.E:** The source radiation pattern is ignored in this assumption since averaging effect of multiple-scattering process ceases its effect on the S-wave coda. However it could be still influential on direct S wave portion in a way by altering attenuation estimates in cases of poor azimuthal coverage of station distribution with respect to earthquakes. In our case, the seismic network used in this study has relatively good azimuthal coverage, and thus direct S-wave measurements averaged over a lot of stations vanish influence of source radiation pattern on the attenuation estimates. The source radiation patterns, for example, will probably have a minor effect on our results because in the late coda the effect vanishes due to the averaging by multiple scattering and the direct S-wave is measured and averaged over a lot of stations.

[revised manuscript text omitted]